# Invariant representations of mass in the human brain

Sarah Schwettmann[1,2,3]*, Joshua B Tenenbaum[1,2,3,4], Nancy Kanwisher[1,2,3]

[1]Department of Brain and Cognitive Sciences, Massachusetts Institute of Technology, Cambridge, United States; [2]Center for Brains, Minds, and Machines, Massachusetts Institute of Technology, Cambridge, United States; [3]McGovern Institute for Brain Research, Massachusetts Institute of Technology, Cambridge, United States; [4]Computer Science and Artificial Intelligence Laboratory, Massachusetts Institute of Technology, Cambridge, United States

**Abstract** An intuitive understanding of physical objects and events is critical for successfully interacting with the world. Does the brain achieve this understanding by running simulations in a mental physics engine, which represents variables such as force and mass, or by analyzing patterns of motion without encoding underlying physical quantities? To investigate, we scanned participants with fMRI while they viewed videos of objects interacting in scenarios indicating their mass. Decoding analyses in brain regions previously implicated in intuitive physical inference revealed mass representations that generalized across variations in scenario, material, friction, and motion energy. These invariant representations were found during tasks without action planning, and tasks focusing on an orthogonal dimension (object color). Our results support an account of physical reasoning where abstract physical variables serve as inputs to a forward model of dynamics, akin to a physics engine, in parietal and frontal cortex.

## Introduction

Engaging with the world requires a model of its physical structure and dynamics – how objects rest on and support each other, how much force would be required to move them, and how they behave when they fall, roll, or collide. This intuitive understanding of physics develops early and in a consistent order in childhood; infants can differentiate liquids from solids by 5 months of age (*Hespos et al., 2009*; *Hespos et al., 2016*), infer an object's weight from its compression of a soft material by 11 months (*Hauf et al., 2012*), and use an object's center of mass to judge its stability on the edge of a surface by 12 months (*Baillargeon, 1998*). By adulthood, human physical reasoning is fast and rich, and it generalizes across diverse real-world scenarios. Yet little is known about the brain basis of intuitive physics, which could enable direct tests of computational models by revealing the relevant neural representations and their invariances and automaticity.

A key distinction between computational models of intuitive physics is whether they use model-free pattern recognition (such as deep neural networks) (*Lerer et al., 2016*), or causal generative models of physical object representations and their dynamics (*Battaglia et al., 2013*). The generative approach models physical reasoning as approximate probabilistic inference over simulations in a physics engine, an architecture with two core parts: an object-based representation of a 3D scene (which encodes many static variables, such as the size and mass of each object), and a model of physical forces that govern the scene's dynamics. These models may make use of deep neural networks, but also contain additional structured information about the world. Unlike the pattern recognition approach, the generative framework entails extraction of abstract representations of physical concepts and laws that support generalization, mirroring the human capacity to reason about novel physical scenarios without training. Within this framework, simulation-based models can make robust

*For correspondence:
schwett@mit.edu

Competing interests: The authors declare that no competing interests exist.

inferences with accuracy comparable to human performance across many areas of physics, including collisions (*Smith et al., 2013a*), (*Smith et al., 2013b*), fluid dynamics (*Bates et al., 2015*), the motion of granular materials (*Kubricht et al., 2016*), and predictions about the outcome of applied forces (*Ullman et al., 2018*; *Hamrick et al., 2016*).

A recent fMRI study has implicated specific regions in the parietal and frontal lobes in intuitive physical inference in humans (*Fischer et al., 2016*). These regions responded more strongly during a physical reasoning task (which direction a tower of blocks will fall) than a difficulty-matched non-physical discrimination performed on the same stimuli, and more strongly during viewing of animated shapes depicting physical interactions of inanimate objects than social interactions of agents (*Fischer et al., 2016*). The candidate regions for intuitive physical inference found in this study resemble regions previously implicated in action planning (*Johansson and Flanagan, 2009*; *Gallivan et al., 2014*; *Evarts and Thach, 1969*; *Loh et al., 2010*; *Chouinard et al., 2005*; *van Nuenen et al., 2012*) and tool use (*Gallivan et al., 2013*; *Brandi et al., 2014*; *Valyear et al., 2012*; *Goldenberg and Hagmann, 1998*; *Goldenberg and Spatt, 2009*), consistent with the importance of physical understanding for these functions (*Cant and Xu, 2012*). However, crucially, it is unknown what these regions represent about physical events. A pattern recognition approach to physical reasoning might predict that the neural representations in these regions would hold information about low-level visual features or situation-specific representations of physical variables. In contrast, if these regions support a generalized engine for physical simulation, we would expect to find that they hold representations of abstract physical dimensions that generalize across scenario and other physical dimensions.

To answer this question, we conducted three experiments using fMRI to test the generalizability and automaticity of neural representations of a key variable underlying physical reasoning: mass. Mass is not the only physical variable of interest, but it is the most basic scalar quantity that captures a property of all objects and that governs motion in every physical interaction, via Newton's second law. Hence it is a natural first candidate to probe representationally in neural circuitry putatively instantiating a mental physics engine. Participants were scanned with fMRI while performing physical inference, prediction, and orthogonal tasks on visually-presented stimuli. Each scanning session began with two runs of a previously developed 'localizer' task (*Figure 1a*) to identify in each subject individually candidate regions engaged in physical reasoning (*Fischer et al., 2016*). We then we applied pattern classification methods to fMRI data obtained from subjects viewing videos of dynamic objects, to test for invariant representations of mass in these regions (*Fischer et al., 2016*), as predicted if they implement a causal generative model of the physical world.

## Results

### Experiment 1: mass inference

We began by asking whether object mass could be decoded from neural activity in previously-described (*Fischer et al., 2016*) candidate physics regions while participants performed a mass inference task. Six subjects were scanned using fMRI while viewing 3 s movies of real objects interacting in various physical scenarios: splashing into a container of water, being blown across a flat surface by a hairdryer, and falling onto the soft surface of a pillow (*Figure 1b*). Three rigid 3D shapes of equal volume were used (a rectangular prism, a cone, and a half-sphere), and movies were filmed for two different colors and two different masses (45 g, 90 g) of each shape (36 total movies). Visual cues from the scene could be used to infer the mass of each object, which was never explicitly stated. After each movie, subjects responded to a text prompt ('Light or Heavy?') with a button press indicating their inferred mass (*Figure 1c*). Accuracy on this task was 88% (i.e., percentage of responses matching the ground truth outcome) across six subjects.

We first identified the set of parietal and frontal voxels implicated in physical inference in each subject individually using the localizer task (see Materials and methods). We then applied multivariate decoding analyses to fMRI responses in the main experiment to each stimulus of each voxel in that set. To test for situation-invariant mass decoding, linear SVMs were trained on the responses to two of the scenarios (e.g., 'splash' and 'blow'), and tested on the third ('pillow'). This situation-invariant decoding was significant in the candidate physics system, with a group mean accuracy of 0.64 ($p=0.0304$, two-sided t-test, t-statistic = 2.9913, df = 5, significance threshold = 0.05). Critically, this

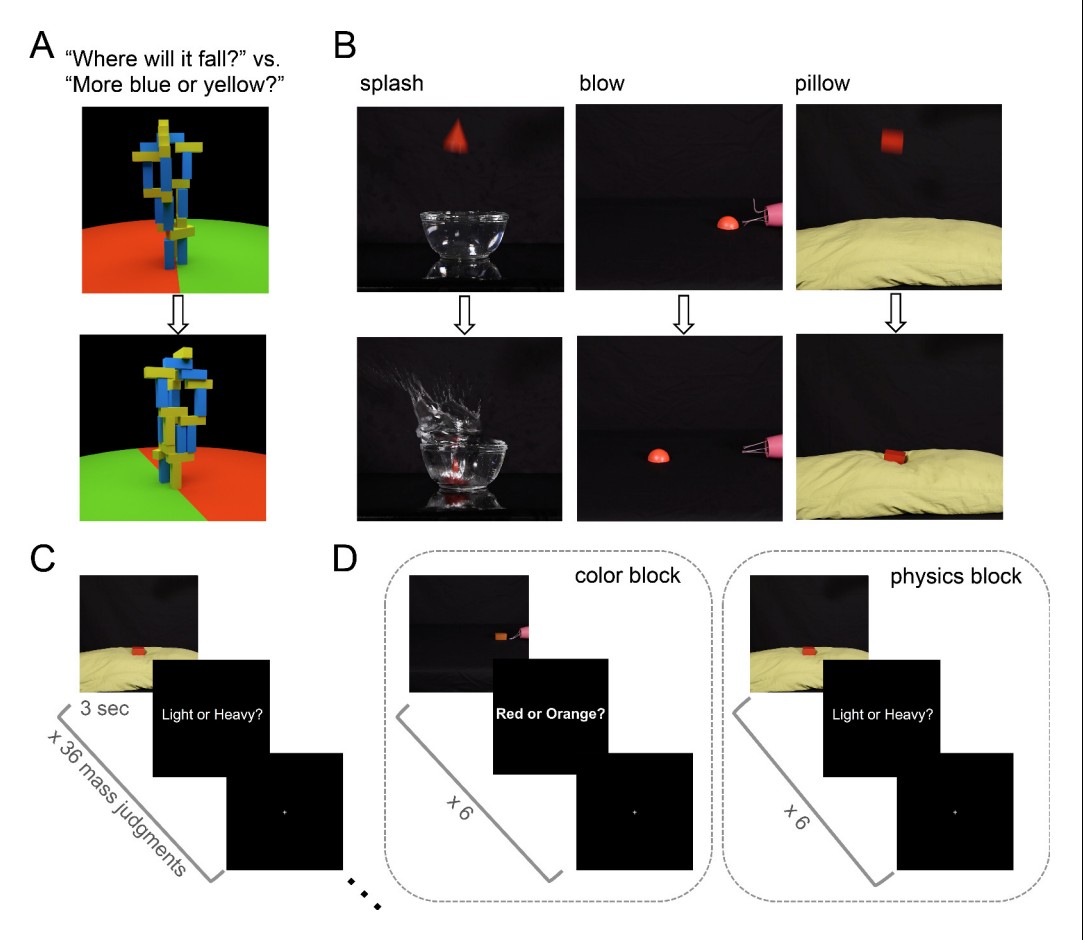

**Figure 1.** Stimuli and tasks from Experiments 1 and 2. (a) Toppling tower task (adapted from *Fischer et al., 2016*) used as a localizer for all experiments. Still frames show an example tower from two different viewpoints during the 360° pan video. Participants were asked in different blocks to determine which side the tower would fall toward (red versus green), or whether the stimulus contained more blue or yellow blocks. (b) Stills extracted from example mass inference videos used Experiments 1 and 2 (top is extracted from early in video, bottom from later). Stills from 'splash' and 'pillow' videos show a heavy object; stills from the 'blow' condition depict a light object. (c) Event-related scanning paradigm in Experiment 1. Each run (4 per subject) presented 36 videos in randomized order (144 total trials with each video presented 4 times), each followed by a 1 s response period ('Light or Heavy?') then a rest period of variable duration (mean 6 s). (d) Experiment 2 used a block design to compare decoding during physics and color blocks. Each run (6 per subject) consisted of 5 color blocks, 5 physics blocks, and 4 (12s) rest blocks. 6 videos were shown in each block (360 total trials with each video presented 5 times in a physics block and 5 times in a color block).

representation of object mass does not depend on whether the object is splashing into water, being blown by a hair dryer, or being dropped onto a pillow. Mean classification accuracies across all three scenarios as well as classification accuracies for each left out scenario individually were greater than 50% in each subject. Further, mass representations are not confounded with shape or color, as colors and shapes were represented in equal proportions for both masses in the training and testing data. Decoding could also not be based on the amount of motion in the videos, as heavy objects caused more motion in two of the conditions (splash, pillow), but did not move at all in the third (blow; see Materials and methods). Finally, decoding could not be based on specific motor responses, because the assignment of buttons to responses was switched halfway through the experiment, with equal an equal number of trials per button-press-to-response assignment represented in training and testing data.

## Experiment 2: mass decoding during color judgment

Experiment 1 suggests we can decode an abstract, generalizable representation of mass from candidate physics regions, but two questions remain. First, does mass encoding occur even if not required by the task? Second, an alternative account of our apparent ability to decode object mass is that we may be decoding instead a prepared response to the explicit mass task ('Light or Heavy?'), which is constant across scenarios. Note that our mass decoding could not simply reflect decoding of a literal motor plan, as the assignment of response meanings to button presses was switched halfway through the experiment, but the hypothesis remains that in Experiment 1 we were decoding an abstract response code invariant to the specific motor plan it would later be translated into. To test this hypothesis, as well as the automaticity of the mass representation, we used a design that interleaves blocks of the mass task and a color task on the same stimuli. This design enabled us to ask whether a situation-invariant mass representation can also be decoded from multivoxel activity during blocks where subjects perform the orthogonal color task where mass was not relevant. Subjects viewed the same stimuli used in Experiment 1, and were prompted both at the beginning of each block and after each video to respond whether the object was 'Light or Heavy?' or 'Red or Orange?' (*Figure 1d*).

In six new subjects, we replicated the findings of Experiment 1: mean situation-invariant decoding accuracy of 0.63 (across scenarios) was significantly above chance (p=0.0357, two-sided t-test, t-statistic = 2.853, df = 5), and decoding was found in 6 out of 6 subjects individually during the mass task (task accuracy 87%). More importantly, mass decoding was also significantly above chance (mean = 0.61, p=0.0033, two-sided t-test, t-statistic = 5.2576, df = 5), and present in each subject individually, during the color task. This result shows that mass is represented even when the task does not require it, and further that the decoding of mass we observe cannot be explained as an abstract response code. Further evidence against the idea that the mass representations reflect response codes comes from the fact that color decoding from the same voxel activity during the color task was at chance in all subjects. Thus the candidate physics engine does not represent all task-relevant dimensions and may be more specific to physical variables.

However, the results of Experiment 2 do leave open the possibility that a context effect from the mass blocks carried over to and created biases on color blocks, contributing to mass decoding during the color task. This motivated our design of a third experiment to test mass decoding in an experiment where subjects were never asked about mass.

## Experiment 3: physical prediction in a collision task

We asked in Experiment 3 whether mass could be decoded from candidate physics brain regions during a physical prediction task that requires mass knowledge but never explicitly interrogates it. We created 48 real-world movies. Each 6 s video shows an object (made of aluminum, cardboard, lego, or cork) sliding down a ramp and colliding with a puck (half-ping-pong ball), whose initial location is consistent between videos (*Figure 2b*). In the task, subjects answer (as immediately as possible) whether they predict the sliding object will launch the puck across a black line, which can lie in three different locations. The mass of the object and its coefficient of friction determine how far it will launch the puck. Importantly, these stimuli were designed in a way that orthogonalizes mass, friction, and motion in the videos (*Figure 2a*), allowing us to test whether it is possible to decode a generalized representation of mass invariant to friction and motion. Each of the four different materials was used to make two objects, a 2.5" cube and a 2.5"x 2.5"x1.25" object with half of the volume of the cube and the same surface area in contact with the ramp. Materials were chosen with densities such that same-volume objects made out of aluminum and cardboard have the same mass (15 g for the small volume 30 g for the large volume), and same-volume objects made from lego and cork have the same mass (45 g or 90 g), while pairs along the other invariance dimension (aluminum and legos, cardboard and cork) share similar coefficients of friction with the ramp (aluminum: $\mu_k$=0.21, lego: $\mu_k$=0.22; cardboard: $\mu_k$=0.40, cork: $\mu_k$=0.46). Accuracy in the prediction task was 71% across 20 subjects.

Experiment 3 replicated once again our finding that mass can be decoded from candidate physics regions (mean accuracy of 0.60 was significant, p=0.0392, two-sided t-test, t-statistic = 2.2152, df = 19, significance threshold = 0.05 ). Further, this experiment demonstrates an important new invariance of these mass representations beyond those already found in Experiments 1 and 2: the

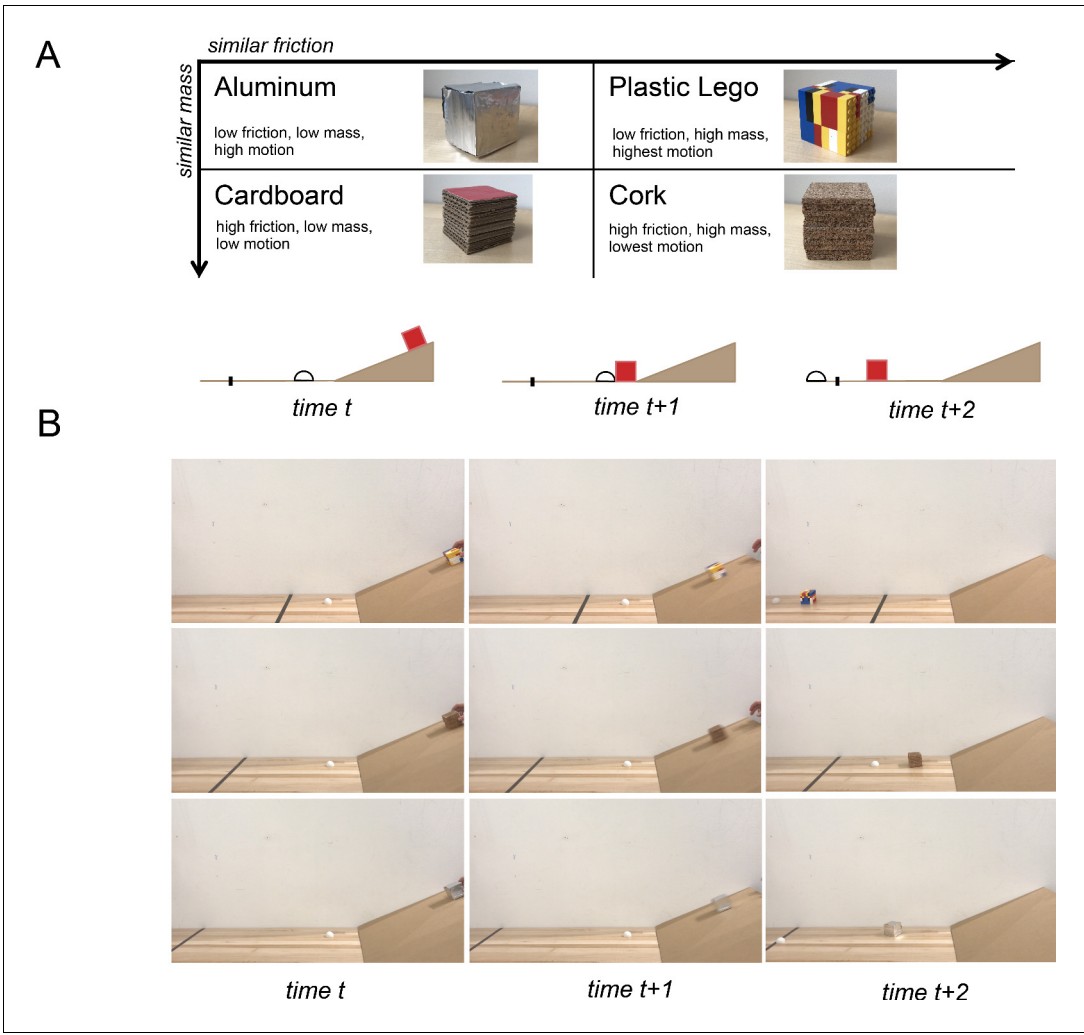

**Figure 2.** Experiment 3 design. (a) Schematic of stimulus design and ramp scenario. To test the invariance of the mass representation to other physical dimensions, this design was chosen to unconfound mass from dimensions of friction, motion, and material (though it was not possible to unconfound these dimensions from each other). (b) Still frames from stimulus videos with examples of 3 material types and 3 possible line locations. Rows (1: lego, 2: cork, 3: aluminum) represent individual videos.

mass decoding in Experiment 3 required generalization across the friction and material of the object shown (lego to cork for heavy, and cardboard to aluminum for light), as well as generalization across the amount of motion in the videos (calculated by measuring the amount of optical flow; see Materials and methods). To minimize the difference in eye movements across trials, participants were instructed to fixate on a black cross in the center of the screen during each video. Eye movements were recorded for 6 subjects, to verify that subjects were fixating and to ensure that mass decoding was independent of eye movement (see Materials and methods). A two-way ANOVA with mass and friction as repeated-measures factors revealed no significant effects at the .05 significance level of mass or friction on mean eye position (mass: $F_{1,3} = 0.084$, p=0.79; friction: $F_{1,3} = 0.31$, p=0.62) or mean saccade amplitude (mass: $F_{1,3} = 0.28$, p=0.63; friction: $F_{1,3} = 0.46$, p=0.55), so it is unlikely that eye movements could explain our decoding results.

We next tested whether object mass could be decoded from regions beyond candidate physics fROIs, namely, regions in the ventral visual pathway outside traditional motor and premotor areas shown to represent object weight during action planning (*Gallivan et al., 2014*). Following Gallivan et al., we used a localizer task (*Cant and Xu, 2012*) based on the contrast of object textures and ensembles versus their scrambled counterparts, to identify LO and texture-sensitive regions of OTC

in six participants completing the ramp task in the same session. Although these subjects showed reliable decoding of a mass representation invariant to friction, material, and motion in candidate physics regions during the physical prediction task (Experiment 3), this invariant decoding did not reach significance in LO (p=0.39) or in OTC (p=0.67) during the same task. While the parietal and frontal regions previously implicated in intuitive physics are recruited to compute an abstract representation of mass useful in a generalized model of physics, regions in the ventral stream, canonically associated with visual pattern recognition, may be recruited to infer object mass tied to scene- and task-specific cues such as the visual appearance of object material.

## Analyses across all experiments

We used all data (two runs per subject) from the toppling towers localizer to perform a whole-brain random-effects group analysis for the physics >color contrast (*Figure 3A*). This group analysis identifies a map of brain regions, primarily premotor and parietal areas, that was first shown in *Fischer et al. (2016)* to be preferentially engaged in physical reasoning, and is now replicated here in 32 new subjects. We further demonstrate that this candidate physics network encodes an abstract, generalizable representation of object mass that can be decoded from individual subject fROIs (see Materials and methods) in 31 out of 32 subjects (*Figure 3B*).

## Discussion

In a network of parietal and frontal brain regions previously implicated in intuitive physical inference, and replicated here in a larger sample (see *Figure 3*), we find robust decoding of object mass, replicated across three experiments and present numerically in 31 out of 32 participants. Critically, this mass representation is invariant to the scenario revealing the object's mass (splashing, falling, and blowing), as well as to object material, friction, and motion energy. In everyday physical reasoning, humans are able to use visual cues in a single scene to infer physical properties of an object that can be generalized to predict the object's dynamics in novel scenes, plan actions upon the object, and make inferences about similar but unfamiliar objects. Here we present the first neural evidence of a mass representation underlying physical reasoning with invariance that supports this kind of flexible, generalizable navigation of the physical world. Among current computational models, those that best exhibit this capacity for generalization are structured generative models such as physics engines (*Battaglia et al., 2013*; *Yildirim et al., 2019*), supporting the hypothesis that the network of frontal and parietal fROIs we identify implements in some form a causal generative model of physical objects and their dynamics.

To date, neural representations underlying physical reasoning have only been studied in action planning tasks. *Gallivan et al. (2014)*. used multivariate decoding methods to find, in multivoxel activity patterns during action planning, representations of object mass in ventral visual pathway areas, specifically the lateral occipital complex (LO), posterior fusiform sulcus (pFs), and texture-sensitive regions of occipitotemporal cortex (OTC), in addition to motor cortex (M1) and dorsal premotor cortex (PMd), where mass information for action planning is known to be represented (*Johansson and Flanagan, 2009*; *Evarts and Thach, 1969*; *Loh et al., 2010*; *Chouinard et al., 2005*; *van Nuenen et al., 2012*). Our work goes beyond prior studies reporting neural decoding of mass in two key respects. First, prior studies have provided evidence for representations of mass (*Johansson and Flanagan, 2009*; *Gallivan et al., 2014*; *Evarts and Thach, 1969*; *Loh et al., 2010*; *Chouinard et al., 2005*; *van Nuenen et al., 2012*) only when participants are performing action planning tasks. In contrast, we show that these representations are available when subjects are not asked about mass per se, for instance in the ramp task where mass is relevant to the task but not explicitly reported, and in the color task where mass is not relevant at all. Second, and more importantly, prior studies have decoded representations of mass only within a particular stimulus or scenario, whereas our study finds abstract representations of mass that generalize across scenarios and are invariant to friction, material, and motion. It is the abstractness and invariance of the mass representations reported here that suggests they reflect not just another dimension of visual pattern classification, but the generalizability expected of inputs to a physics engine in the brain.

These invariant representations of mass are found in a network of frontal and parietal regions (*Figure 3*) that we suggest support machinery for a neural physics engine. Similar frontal and parietal regions have been implicated in thinking about physical concepts presented as words (*Mason and*

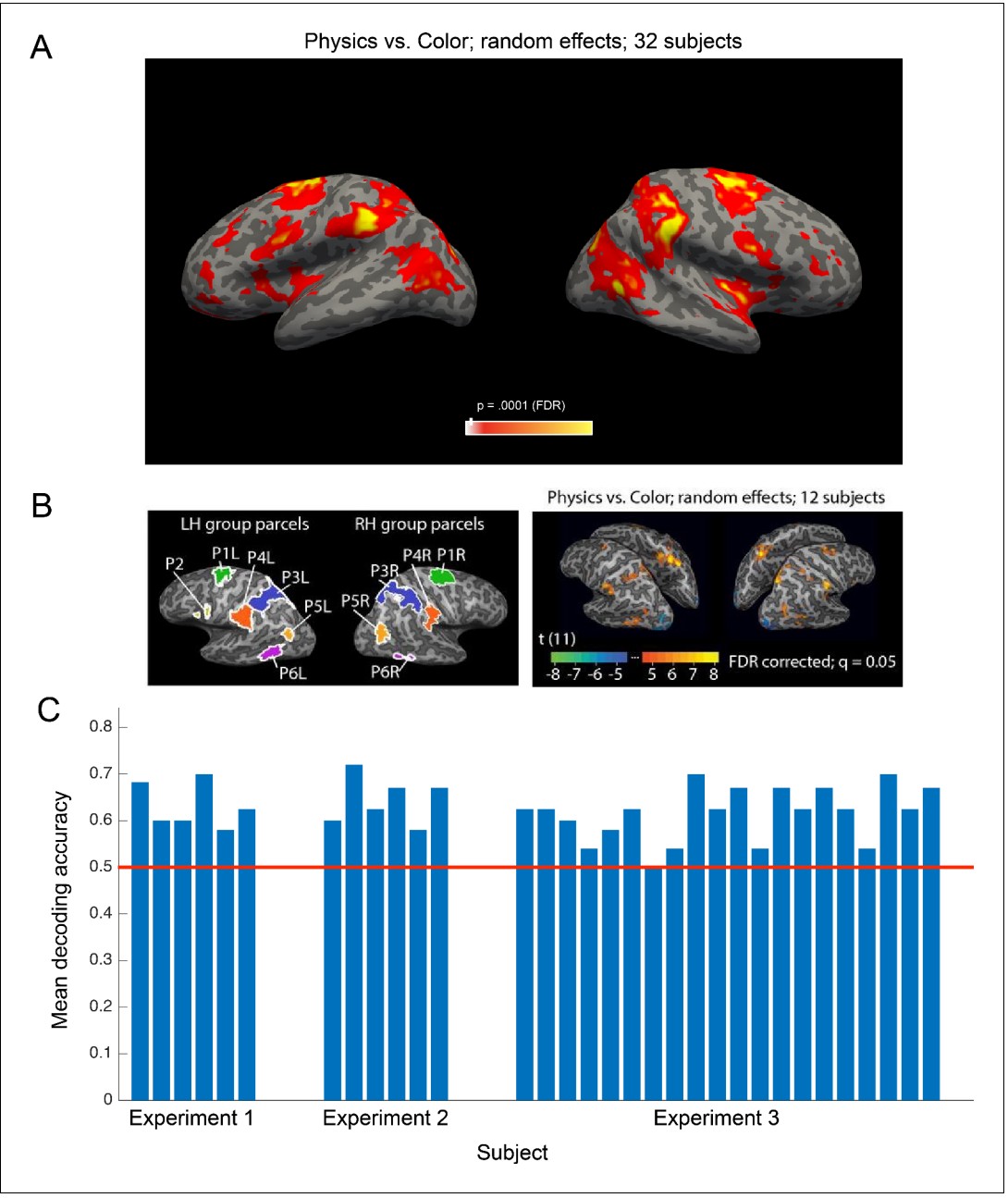

**Figure 3.** Main findings from all participants in all experiments. (a) Group random effects map for the physics >color contrast in the localizer task based on all subjects (*n* = 32, two runs per subject p<0.0001 FDR), replicating the pattern reported by *Fischer et al. (2016)*. (b) Group parcels and random effects map from all subjects in *Fischer et al. (2016)*. Group parcels for the physics >color contrast computed using one run per subject (*n* = 12; left-out data from the other run used for validation); random effects map for the physics >color contrast based on all data (two runs per subject). Significant voxels in the group random effects analysis generally fall within the parcels identified in the parcel-based analysis, but not necessarily vice versa (the random effects map may underestimate the extent of the cortex engaged by the task due to anatomical variability across subjects). (c) Mean accuracies decoding mass from candidate physics fROIs in each subject across the three experiments. Decoding analyses were carried out on data from all parcels. A two-way ANOVA did not reveal a significant effect of L or R hemisphere (p=0.54) or frontal or parietal parcel (p=0.86) on decoding accuracy.

*Just, 2016*), supporting the hypothesis that this network represents abstract, generalizable physical concepts rather than low-level visual features or situation-specific representations of physical variables. We did not find invariant mass representations in ventral visual pathway areas such as LOC and OTC (in tasks not requiring action planning), suggesting that LOC may not play a causal role in computing object weight. This supports previous findings by *Buckingham et al. (2018)*, which showed that a patient with bilateral lesions including LOC had a preserved ability to judge object weight (*Buckingham et al., 2018*). This overarching pattern of results suggests that when ventral visual areas do represent motor-relevant object properties (*Gallivan et al., 2014*), it may be a top-down effect driven by the motor planning process where representations are tied to specific tasks.

It remains unknown how the brain estimates generalizable physical properties of objects from visual inputs. It could be that feed-forward inference methods, akin to deep-learning based recognition models, are integrated with generative models and provide an efficient means of inference of physical properties that then serve as inputs to physics engines. This account has been explored computationally and has received behavioral support (*Yildirim et al., 2019*), but how such a model may be instantiated between frontal and parietal regions underlying generalized physical reasoning and traditional object-driven cortex is an open area of investigation. Our results show that at the level of prediction and inference, intuitive physics recruits brain regions and representations outside the ventral stream, the canonical locus of visual pattern recognition.

Our findings open up numerous avenues for further investigation. Mass is just one of the properties underlying intuitive physical reasoning. Future investigations can test whether the same or other brain regions represent other physical dimensions, types of physical forces and events, and domains outside of rigid body physics (e.g. the viscosity of liquids and the restitution of materials). How fine-grained is the neural representation of mass? Future work should also test the relationship between the amount of variance in real world physical properties, and the fine-grainedness of their neural representations. Do the same neural representations that underlie physical inference also underlie action planning (*Johansson and Flanagan, 2009*; *Loh et al., 2010*; *Chouinard et al., 2005*; *van Nuenen et al., 2012*)? A model-based account of physics in the brain could support both physical inference and action planning in the same underlying brain regions, which may serve as the seat of a neural physics engine. These studies and others can be expected to shed more light on how the frontal and parietal physics network examined here implements a causal generative model of objects and their dynamics.

We have shown evidence that object mass invariant to physical scenario, friction, object material, and motion, is represented in premotor and parietal brain regions during physical inference and prediction tasks not requiring action. The invariant representation in areas traditionally associated with action during a perceptual judgment suggests that these regions support the type of representation that would serve as the input to a generalized physics engine, useful in understanding forces, dynamics, and even planning actions.

## Materials and methods

### Participants

Six subjects (ages 21–26; 3 male, 3 female) participated in Experiment 1, six (ages 21–40; 3 male, 3 female) in Experiment 2, and twenty (ages 20–32; 9 male, 11 female) in Experiment 3. A power analysis was used to calculate the appropriate number of subjects for each experiment ($p_0 = 0.5$, $p_1 = 1$, $\alpha = 0.01$, desired power = 0.9, one-sided binomial). All participants were right-handed and had normal or corrected to normal vision. All participants provided informed consent before participation. The Massachusetts Institute of Technology Institutional Review Board approved all experimental protocols.

### Experimental design

In each experiment, participants performed 2 runs of a 7 min 'localizer' fMRI task from *Fischer et al. (2016)*, in which subjects viewed 6 s movies depicting stacks ('towers') of yellow, blue, and white blocks created in Blender 2.70 (Blender Foundation). The block towers were constructed to be unstable such that they would topple if gravity were to take effect. Each tower was positioned in the center of a floor where half of the floor was colored green, and the other half red. In each movie the

tower itself remained stationary while the camera viewpoint completed one 360° pan, providing a range of vantage points of the tower. While viewing each movie, subjects were instructed to perform one of two tasks: imagine how the blocks would fall and report whether more blocks would come to rest on the red or green side of the floor (physics task), or report whether there are more blue or yellow blocks in the tower (color task). A physics > color contrast was used to identify candidate physics functional ROIs (fROIs) in each subject individually (see below) within which decoding analyses were subsequently performed.

Each scanning run for this localizer task (2 per subject) consisted of 23 18 s blocks: 10 blocks of the physical task, 10 blocks of the color task, and 3 rest blocks, which consisted of a black screen with a fixation cross. Each nonrest block began with a text cue, displayed for 1 s, which read either 'more blue or yellow?' (color task) or 'where will it fall?' (physics task). The text cue was followed by the presentation of a tower movie (6 s) and then a black screen during a 2 s response period. This sequence was repeated twice within a block, with the same task being cued for both movie presentations within a block.

In the same scanning session, participants performed 4 to 6 runs of the main experimental paradigm, which was different for each experiment, as described below. Each scanning session lasted 2 hr.

## Experiment 1: mass inference

Subjects viewed 3 s video stimuli (*Figure 1*) of three different geometric solids interacting in various visual scenes (splashing into water, falling onto a pillow, being blown across a flat surface) that indicated their mass. In an event-related design, each run (four per subject) presented 36 videos in randomized order, each followed by a 1 s response period, then a rest period of variable duration (mean 6 s). During the response period, subjects were instructed to press a button indicating whether the object they saw was light or heavy. After the button press, no feedback was given to participants on correctness. In all three experiments, the assignment of buttons to responses was switched halfway through the experiment, with an equal number of trials per button-press-to-response assignment represented in training and testing data, to ensure mass decoding could not be based on specific motor responses.

Objects were constructed by hand from Learning Resources 'View-Thru Geometric Solids.' Three shapes of equal volume were selected as stimuli: a cone, half-sphere, and rectangular prism. Two different masses were created for each object: the 'light' objects were left empty (45 g), and the 'heavy' objects were filled with a mixture of lead pellets and flour (90 g), and painted the same color. The visual appearance of the objects was identical across masses, only the object's physical behavior could be used to infer its mass. To create objects of different colors, Adobe Premiere Pro software (Adobe Systems) was used to color-shift the object surface from red to orange, for a total of 36 video stimuli. Decoding analyses in Experiment 1 collapsed across color.

The objects were filmed interacting in three different visual scenarios. Physical parameters of the scene besides object identity and mass were held constant across videos; that is the height from which objects were dropped (splashing, dropping scenarios), the volume of water into which they fell (splashing scenario), or the distance from the hairdryer (air source) at which they were placed (blowing scenario). While the 3D shapes of the objects represented familiar visual forms, the scenarios were selected as novel domains for mass inference. Subjects did not interact physically with the objects before the scan.

## Experiment 2: mass decoding during color judgment

The same stimuli from Experiment 1 were used in Experiment 2. However, in Experiment 2, participants performed a color task in addition to the mass task on the same objects. The two judgment types were matched for difficulty using data collected from 50 workers with normal color vision on Amazon Mechanical Turk. Each worker performed the light/heavy mass task as well as the red/orange color task for all 36 movies. Mean accuracy on the mass task was 86.2% (±2.4 SD), mean accuracy on the color task was 89% (±3.6 SD). During the scanning session, mass and color trials were presented in blocks of 6 trials each. After each video, participants were asked to press a button indicating whether the object was 'Light or Heavy?' (mass task), or 'Red or Orange?' (color task).

Each run (6 per subject) consisted of 5 color blocks, 5 physics blocks, and 4 (12s) rest blocks. Participants did not receive feedback on accuracy.

## Experiment 3: physical prediction in a collision task

In Experiment 3, participants viewed 6 s videos (*Figure 2*) of physical objects sliding down a ramp and colliding with a puck (half ping-pong ball) placed the same distance from the ramp in each video. In an event-related design, each run (4 per subject) presented 24 of the 48 videos in randomized order (subjects saw every video twice in total), each followed by a rest period of variable duration (mean 6 s). Before the experiment, subjects were instructed to respond with a button press, as early as they could within each video, whether they predicted the sliding object would launch the puck across a black line. In each video the line could lie in one of 3 different locations, to discourage memorization of outcome by object. Each run contained equal numbers of each line position (8 trials). After the button press, no feedback was given to participants on correctness. To ensure familiarity with the visual appearance of the objects in the videos and their material properties, subjects were exposed to the physical objects before the scan. All objects were placed on a flat surface and subjects were instructed to 'interact' with each for 5 s. This instruction was chosen instead of 'lift' or 'pick up' to avoid priming attention to mass.

Object materials were selected to orthogonalize mass and friction, object material, and motion. Coefficients of friction were found by taking of the tangent of the angle of incline at which the object starts to slide down the ramp at constant speed, after being tapped. Motion in the videos was calculated using the Optical Flow package in Matlab 2016. Optical Flow identifies moving objects and calculates the amount of motion between video frames to determine the overall amount of motion in x and y dimensions in each video. The most motion was found in the movies with lego blocks (x = 1.1848 e+04, y = 1.8065e+04), followed by aluminum (x = 1.0781e+04, y = 1.767e+04), cardboard (x = 9.6789e+03, y = 1.4150e+04), and cork (x = 9.0324e+03, y = 1.4126e+04).

## Data Acquisition

Imaging was performed at the Athinoula A. Martinos Imaging Center at MIT on a Siemens 3T MAGNETOM Tim Trio Scanner with a 32-channel head coil. A high-resolution T1-weighted anatomical image (MPRAGE) was also collected for each subject (TR = 2.53 s; TE = 1.64, 3.5, 5.36, and 7.22 ms; $\alpha$ = 7°; FOV = 256 mm; matrix = 256 × 256; slice thickness = 1 mm; 176 slices; acceleration factor = 3; 32 reference lines). Whole-brain functional data were collected using a T2*-weighted echo planar imaging pulse sequence (TR = 2 s; TE = 30 ms; flip angle-$\alpha$=90°; FOV = 200 mm; matrix = 64 × 64 mm; slice thickness = 3 mm isotropic; voxel size = 3 × 3 mm inplane; slice gap = 0.6 mm; 32 slices).

## Eye movement recordings

We recorded eye movements (n = 6) with the EyeLink 1000 Eye-Tracker (SR Research) in the scanner. Eye tracking data were preprocessed with EyeLink Data Viewer software and analyzed in MATLAB R2016B (The MathWorks). Data were analyzed to confirm eye movements could not explain mass decoding results. Trials were labeled as light or heavy and low or high friction according to real-world video identity. For each trial, the entire duration of the video (6 s) was used for analysis. Mean eye position (deviation from center of the screen) and mean saccade amplitude (averaging over all saccades that occurred in that trial were calculated. We then used a two-way ANOVA to analyze the interaction between mass and friction and mean eye position and saccade amplitude during the fixation condition and found no significant effects.

## fMRI data preprocessing

Data preprocessing and general linear models were performed using FsFast tools in the FreeSurfer Software Suite (freesurfer.net). All other analyses were conducted in MATLAB R2016b (The MathWorks). Preprocessing consisted of 3D motion correction, slice scan time correction, high-pass filtering via a general linear model with a Fourier basis set (cutoff of two cycles per run, which also achieved linear trend removal), and spatial smoothing with a 4 mm FWHM Gaussian kernel. Before spatial smoothing, the functional runs were individually coregistered to the subject's $T_1$-weighted anatomical image. All individual analyses were performed in each subject's native volume. For group-level analyses, data were coregistered to standard anatomic coordinates using the Freesurfer

FSAverage template. General linear models included 12 nuisance regressors based on the motion estimates generated from the 3D motion correction: *x*, *y*, and *z* translation; *x*, *y*, and *z* rotation; and the approximated first derivatives of each of these motion estimates.

## Group analysis

To test whether a systematic network of regions across subjects responded more strongly to physical judgments than to color judgments in the localizer task, we performed a surface-based random-effects group analysis across all subjects using Freesurfer. We first projected the contrast difference maps for each subject onto the cortical surface, and then transformed them to a common space (the Freesurfer fsaverage template surface). The random-effects group analysis yielded a network of brain regions (p<0.0001) preferentially engaged in physical reasoning that replicated the pattern reported by *Fischer et al. (2016)*.

## fROI Definition

To examine the information represented in candidate brain regions for physical inference, we defined functional regions of interest (fROIs) in each subject individually by intersecting subject specific contrast maps with group-level parcels. Following *Fischer et al. (2016)*, we used the towers localizer to identify brain regions in each subject that displayed a stronger response to the physics task than to the color task. These individual subject maps were then intersected with group-level physics parcels identified in *Fischer et al. (2016)* that were shown to be preferentially engaged in physical reasoning. Specifically, Fischer first identified 11 group-level parcels from the physics >color contrast on toppling tower stimuli (*Figure 3b*). Fischer et al. suggest that the spatial content of the physics task (not present in the color task, as individual block positions were irrelevant) may have contributed to responses in candidate physics regions. A second experiment was used to control for task differences, where physical and social prediction tasks were contrasted on the same set of moving dot stimuli. In this experiment, subjects watched pairs of moving dots with motion implying social interaction (like classic Heider and Simmel animations) or physical interaction (like billiard balls). In each video, one of the dots disappeared and subjects were asked to predict its trajectory. Both conditions required mental simulation of spatial paths, but one implicitly invoked physical prediction and the other implicitly invoked social prediction. Only a subset of the parcels showed a significantly greater response to physical vs. social interactions: P1L and P1R (bilateral parcels in dorsal premotor cortex and supplementary motor area), P3L and P3R (bilateral parcels in parietal cortex situated in somatosensory association cortex and the superior parietal lobule, and P4L (the left supramarginal gyrus). We found individual subject fROIs by intersecting subject data from the physics >color contrast with these five parcels (in volumetric space for each subject), retaining only the voxels that fell within the intersection. In this way, fROI locations were allowed to vary across individuals but required to fall within the same parcel to be labeled as a common ROI across subjects. Subsequent decoding analyses were performed in individual subject fROIs.

## Decoding analysis

To test the representational content of multivoxel activity from candidate physics regions, decoding analyses (*Naselaris et al., 2011*; *Haxby et al., 2014*) were run on multivoxel activity across voxels in these fROIs. An SVM was used for classification, restricted to linearly decodable signal under the assumption that a linear kernel implements a plausible readout mechanism for downstream neurons (*Shamir and Sompolinsky, 2006*; *DiCarlo and Cox, 2007*). In each of 3 experiments we tested the invariance of physical representations by testing the classifier on data from conditions that differed from those in the data used for training along a key dimension. Trials were classified for decoding based on actual trial identity (whether the object was light or heavy). Only the data from the 3 s video was included in the decoding analysis, the 1 s response period (Experiments 1 and 2) was not used for decoding. A canonical HRF response was assumed, with the HRF aligned to the start of the video. To decode mass in Experiment 1, an SVM was trained on beta values (from all voxels within individually-defined fROIs) classified as corresponding to either 'heavy' or 'light' conditions, collapsing across shape and color. We used two of the three scenario types (splash, blow, pillow) to train the classifier and tested on the third, left-out scenario, forcing the classifier to generalize across physical scenarios and iterating over left-out conditions to obtain a mean classification accuracy for

each subject. Correction for multiple comparisons was not performed, given independent data for each subject and repeated replication in multiple individual subjects. In Experiment 2, the same procedure was used to decode mass during both mass and color tasks in the interleaved block design, thus testing (i) whether Experiment 1 replicates and mass can be decoded during the mass task, and (ii) whether mass representations can be decoded from candidate physics regions during an irrelevant (color) task.

We used similar multivariate analyses to test whether we could decode mass from candidate physics fROIs during the physical prediction task in Experiment 3. Experiment 3 used an event-related design where trials were 6 s videos of objects sliding down a ramp. Decoding analyses were done on data from the entire video, with HRFs aligned to video onset. To test decoding of mass invariant to friction and motion, we trained an SVM on beta values from two conditions that differ in the mass dimension but not in friction or size (e.g. light, low friction versus heavy, low friction), and tested on the left out conditions (e.g. light, high friction versus heavy, high friction) thus forcing the classifier to generalize across coefficients of friction. This procedure was iterated over left-out conditions to obtain a mean classification accuracy. This decoding of mass is also invariant to material, as objects in the training conditions (e.g. aluminum, legos) have different material composition than objects in the testing conditions (e.g. cardboard, cork).

## Acknowledgements

We thank J Fischer for helpful discussion around experimental design and analysis, L Isik for helpful discussion around MVPA decoding, and A Takahashi and S Shannon for assistance with fMRI scanning at the Athinoula A Martinos Imaging Center at MIT. This work was supported by the National Science Foundation Science and Technology Center for Brains, Minds, and Machines, and the Office of Naval Research Multidisciplinary University Research Initiatives Program (ONR MURI N00014-13-1-0333).

## Additional information

### Funding

| Funder | Grant reference number | Author |
|---|---|---|
| National Institutes of Health | Grant DP1HD091947 | Nancy Kanwisher |
| National Science Foundation | Science and Technology Center for Brains, Minds and Machines | Joshua B Tenenbaum Nancy Kanwisher |
| Office of Naval Research | Multidisciplinary University Research Initiatives Program ONR MURI N00014-13-1-0333 | Joshua B Tenenbaum Nancy Kanwisher |

The funders had no role in study design, data collection and interpretation, or the decision to submit the work for publication.

### Author contributions

Sarah Schwettmann, Conceptualization, Formal analysis; Joshua B Tenenbaum, Conceptualization, Supervision; Nancy Kanwisher, Conceptualization, Supervision, Funding acquisition

### Author ORCIDs

Sarah Schwettmann (iD) https://orcid.org/0000-0001-6385-1396

### Ethics

Human subjects: All participants provided informed consent before participation. The Massachusetts Institute of Technology Institutional Review Board approved all experimental protocols (protocol number: 0403000096).

Decision letter and Author response
Decision letter https://doi.org/10.7554/eLife.46619.sa1
Author response https://doi.org/10.7554/eLife.46619.sa2

## Additional files

### Supplementary files
- Transparent reporting form

### Data availability
All data collected in this study is available on OpenNeuro under the accession number 002355 (https://openneuro.org/datasets/ds002355/).

The following dataset was generated:

| Author(s) | Year | Dataset title | Dataset URL | Database and Identifier |
|---|---|---|---|---|
| Sarah Schwettmann, Joshua B Tenenbaum, Nancy Kanwisher | 2019 | Intuitive physics with fMRI | https://doi.org/10.18112/openneuro.ds002355.v1.0.0 | OpenNeuro, 10.18112/openneuro.ds002355.v1.0.0 |

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
