## [Decision Letter]

**Acceptance summary:**

The authors provide evidence that humans possess a constellation of brain regions involved in encoding the mass of viewed objects (which they refer to as physical inference), and show that this encoding of mass generalizes across objects and various material and physical properties. The reviewers and editors felt that the study was interesting and novel. The experiments and analyses appear to be expertly conducted, and the authors provide considerable evidence in support of their claims. Of the few studies that have investigated the neural coding of object properties such as mass and other physical object properties, few if any have demonstrated that physical inferences of this nature arise in tasks that do not require actions. Therefore, we believe this work will be of great interest to the scientific community.

**Decision letter after peer review:**

Thank you for submitting your article "Invariant representations of mass in the human brain" for consideration by *eLife*. Your article has been reviewed by three peer reviewers, one of whom is a member of our Board of Reviewing Editors, and the evaluation has been overseen by Michael Frank as the Senior Editor. The following individuals involved in review of your submission have agreed to reveal their identity: Jacqueline Snow (Reviewer #2); Jason Gallivan (Reviewer #3).

The reviewers have discussed the reviews with one another and the Reviewing Editor has drafted this decision to help you prepare a revised submission.

Summary:

The authors provide evidence that humans possess a constellation of brain regions involved in encoding the mass of viewed objects (which they refer to as physical inference), and show that this encoding of mass generalizes across objects and various material and physical properties. The reviewers and editors felt that the study was interesting and novel. That said, there are significant concerns about several technical and interpretation issues that need to be addressed before the paper can be published. The following comments highlight the "essential revisions" (which the authors must address in the revised manuscript and response letter).

Essential revisions:

1) In general, the description of the decoding analyses did not contain sufficient details for the study to be replicated. The following points should be clarified in the revised manuscript.

A) What cross-validation procedure was used for the analyses: leave-one-run-out or something else? One gets the sense that trial types were reserved for classifier testing, but that these trials would have also appeared in the same experimental run as the testing trials (e.g., In the Experiment 1 decoding). This could inflate the decoding results.

B) It was also not fully clear how decoding was setup for Experiment 1, and how trials were grouped for leave-one-trial-out classification analysis. Were trials labelled as heavy vs. light in the splash and blow conditions and then tested on pillow trials? This should be clarified.

C) Also, in the Materials and methods, the authors mention that beta values were used on trials 'classified' as… – do you mean based on the subject's behavioral responses or the actual trial identity? For example, in Experiment 1 (and other experiments), what were the regressors used? Was the whole 3 seconds modeled as a box? What about the 1s response period? Was reaction time, etc taken into account? Which betas were then used for classification? Was beta estimated for each trial, so classification was done per trial (that would make sense, but just clarifying). We assume a canonical HRF response was assumed? What HRF parameters? Since this is event related design, the hemodynamic response might leak across trials, which might inflate accuracies (e.g., two adjacent trials might be in the training and test sets respectively resulting in information leak). How was this handled in the decoding?

D) We are not sure what's the dimensionality of the input to the SVM. Were the input beta coefficients from all voxels (which would be thousands of voxels) or was it an 11-dimensional vector, corresponding to the beta coefficients of the 11 functional ROIs? If the latter, was the GLM performed for average time courses within each of 11 ROIs? Or GLM was performed for all vertices and then the betas were averaged within each ROI?

E) What regularization parameter was used for the SVM? How was this hyperparameter selected?

F) In Experiment 3, it was unclear when exactly (in time) authors were decoding mass from the videos (was an HRF aligned to the onset of the video?), and was this conflated at all with the requirement that subjects respond as quickly as possible during the video whether the object will cross the line? Also, how did subjects report this?

G) Please provide more details on how LO/pF/OTC texture areas were defined.

2) Why not perform the analysis in each functional ROI separately? This would provide greater specificity about the effects and allow specific claims to be made about regions rather than a massive frontoparietaltemporal network. On that note, to demonstrate the specificity of the effects, we would also like to see a whole-brain searchlight decoding analysis. This would also allow some greater specificity of where in the brain they see effects of the color task (presumably V4), or where in the brain they see effects of object material (e.g., aluminum vs. cork, more on this below). The inclusion of a searchlight analysis would significantly bolster observations.

3) Based on the Experiment 2 design, we are not sure you can make any claims about the automaticity of mass encoding. The study interspersed color and mass blocks, and it is difficult to rule out that the context effect on mass blocks carried over to (and created biases on) color blocks. Moreover, we think it would have strengthened their observations to show that a classifier trained on mass blocks can cross-decode mass on color blocks, rather than just decode mass from the color blocks independently (as was done).

4) The authors said that "A power analysis was used to calculate the appropriate number of subjects for each experiment." We assumed this meant that the authors determine a sample size of 20 was necessary for Experiment 3. Why then did the authors examine mass decoding in the LO/pF/OTC texture areas in only 6 subjects? A fair comparison, to demonstrate equivalent power, would be to demonstrate that object material (e.g., aluminum vs. cork) can be decoded using the same regions (LO/pF/OTC).

5) Eye Movement Control analysis. This analysis was completely unclear. How are the different measures computed, what time windows were used for analysis (specific events in a trial?), how were the trials collapsed for analysis? No details were provided making it difficult to interpret the authors' claims.

6) "A pattern recognition approach to physical reasoning might predict that the neural representations in these regions would hold information about low-level visual features or situation-specific representations of physical variables. In contrast, if these regions support a generalized engine for physical simulation, we would expect to find that they hold representations of abstract physical dimensions that generalize across scenario and other physical dimensions." – We are a bit unconvinced about the logic. Isn't it possible for a pattern recognition technique to implicitly encode these abstract physical dimensions, which are then detected by your decoding approach? For example, suppose we train a reinforcement learning (RL) algorithm in a virtual environment to perform certain tasks. Isn't it possible that physical dimensions are implicitly encoded by the RL algorithm even though the RL algorithm does not utilize an explicit physical generative model? In fact, one might argue that the use of MVPA (linear SVM) in Experiment 1 suggests that the physical representation is implicit, so the authors might simply be decoding the pattern recognition approach's implicit representation of mass (or some other physical attribute) in the brain?

7) I understand that the authors relied on a network of brain areas that had been identified in a previous study by Fischer et al., 2016, which were shown to be engaged in a physical reasoning paradigm. It does seem, however, from the description of the localizer task, that the contrast of falling direction (i.e., the 'physics' task) vs. color task, would identify a number of different brain areas, not all of which would have similar functions. As one example, intuitively, it seems that the physics task that involved estimating where the leaning tower of blocks would fall would preferentially activate motion-selective areas (i.e., hV5/MT+) – which have been shown in previous fMRI studies to be activated when observers anticipate motion. The group maps illustrated in Figure 3A do indeed suggest an involvement of such motion-selective cortex, especially in the RH. As such, I wondered how meaningful the results are given that there is such a broad network of areas that is being sampled in the analyses. Is the claim that the network as a whole is operating as a 'physics engine', or are more localized regions within the network accounting for this function more so than others? It is also possible that different regions within the network may be more or less likely to show invariance – for example in real-world scenarios when, presumably, more specific rather than invariant representations would be beneficial to support possible actions? Perhaps identifying the areas that were part of this 'network' (either in the figure or a new table) would help readers to make their own assessment.

---

## [Author Response]

Essential revisions:1) In general, the description of the decoding analyses did not contain sufficient details for the study to be replicated. The following points should be clarified in the revised manuscript.A) What cross-validation procedure was used for the analyses: leave-one-run-out or something else? One gets the sense that trial types were reserved for classifier testing, but that these trials would have also appeared in the same experimental run as the testing trials (e.g., In the Experiment 1 decoding). This could inflate the decoding results.

The SVM decoder was trained on two scenarios and tested on one left-out scenario. To minimize information bleed due to the HRF, we used Optseq to randomize both the trial order and ISI, the latter by adding a jitter (mean duration 6s). Even with the jitter, we appreciate the reminder that the trials may still not be completely independent in the long tail of the HRF. To test the effect on decoding accuracy, we ran the decoding analyses again on half the data: training on trials in one half of the runs in each experiment, and testing on left-out trials in the other half of the runs (so trials in the training and testing data could not possibly be adjacent in the experiment). We saw only a minimal decrease in decoding accuracy, to be expected with a decrease in the amount of data, and results remained significant.

B) It was also not fully clear how decoding was setup for Experiment 1, and how trials were grouped for leave-one-trial-out classification analysis. Were trials labelled as heavy vs. light in the splash and blow conditions and then tested on pillow trials? This should be clarified.

In Experiment 1 analyses, a linear SVM decoder was trained on data from labeled heavy and light trials in two scenarios (e.g. splash*, S* and blow, *B*) then tested on data from a third, left out scenario (e.g. pillow, *P*). We iterated over groupings (train *S,B*test *P*; train *S,P* test *B;* train *P,B* test *S*) to obtain three classification accuracies corresponding to each of the scenarios being left out. We report the mean of those classification accuracies in the paper. We are happy to note that the decoding is above chance in each subject in each separate grouping; in other words, it is not that one train/test direction is carrying the result. This result is now reported in the paper.

C) Also, in the Materials and methods, the authors mention that beta values were used on trials 'classified' as… – do you mean based on the subject's behavioral responses or the actual trial identity? For example, in Experiment 1 (and other experiments), what were the regressors used? Was the whole 3 seconds modeled as a box? What about the 1s response period? Was reaction time, etc taken into account? Which betas were then used for classification? Was beta estimated for each trial, so classification was done per trial (that would make sense, but just clarifying). We assume a canonical HRF response was assumed? What HRF parameters? Since this is event related design, the hemodynamic response might leak across trials, which might inflate accuracies (e.g., two adjacent trials might be in the training and test sets respectively resulting in information leak). How was this handled in the decoding?

Trials were classified for decoding based on actual trial identity (whether the actual object filmed was light or heavy). Only the data from the 3s video was included in the decoding analysis, the 1s response period (Experiments 1 and 2) was not used for decoding. A canonical HRF response was assumed, with the HRF aligned to the start of the video. GLMs included one regressor per stimulus condition, as well as nuisance regressors for motion correction (*x, y, z*), and the approximated first derivatives of each of these motion estimates. The response periods were modeled with their own regressors, but we did not include those betas in the analysis. These points are now clarified in the Materials and methods section.

We recognize that the hemodynamic response leak may have inflated decoding accuracy in an event-related design where training and testing data may have come from adjacent trials in the same run (though trial order was randomized in each run for each subject). This concern is addressed in Essential revision 1A.

D) We are not sure what's the dimensionality of the input to the SVM. Were the input beta coefficients from all voxels (which would be thousands of voxels) or was it an 11-dimensional vector, corresponding to the beta coefficients of the 11 functional ROIs? If the latter, was the GLM performed for average time courses within each of 11 ROIs? Or GLM was performed for all vertices and then the betas were averaged within each ROI?

We used beta values from all voxels within individually-defined fROIs as inputs to a linear SVM. The SVM was trained on input data from multiple runs with labels corresponding to different conditions (beta values for hundreds of voxels per condition; thousands of data points overall). This point is now clarified in the Materials and methods section.

E) What regularization parameter was used for the SVM? How was this hyperparameter selected?

A regularization parameter of 1 was used for the SVM. This parameter is the default value for the box constraint in the Matlab fitcsvm function (Matlab R2019a).

F) In Experiment 3, it was unclear when exactly (in time) authors were decoding mass from the videos (was an HRF aligned to the onset of the video?), and was this conflated at all with the requirement that subjects respond as quickly as possible during the video whether the object will cross the line? Also, how did subjects report this?

Experiment 3 used an event-related design where trials were 6s videos of objects sliding down a ramp. Subjects predicted whether the puck would cross a black line and reported their judgment with a button press during each video. We asked subjects to make their predictions as soon as they could within each video because we wanted their judgments to be made before the object came to rest (revealing whether it crossed the line or not), but we also wanted to show each video in its entirety to provide maximal visual cues about the mass of the object (to be decoded). Decoding analyses were done on data from the entire video, with HRFs aligned to video onset.

G) Please provide more details on how LO/pF/OTC texture areas were defined.

LO, pFs, and texture-sensitive regions of OTC were identified in each subject following Gallivan et al., 2014, using localizer tasks from Cant and Xu, 2012. Two runs of the localizers were performed in the same testing session as the prediction task in Experiment 3 of the present study. The stimuli consisted of full-color objects, object ensembles, and surface textures as well as their phase-scrambled counterparts, with each category containing equal numbers of living and non-living stimuli. In each localizer, a run consisted of four blocks each of intact and scrambled objects, or intact and scrambled object ensembles and textures. Subjects fixated on a cross at the center and pressed a button if they detected a slight spatial jitter, which occurred randomly in 1 out of every 10 images. Object-sensitive ROIs (LO and pFs) were defined by the contrast of Objects > Scrambled Objects, and texture-sensitive regions were defined by the conjunction of the contrasts of Textures > Scrambled and Object Ensembles > Scrambled.

2) Why not perform the analysis in each functional ROI separately? This would provide greater specificity about the effects and allow specific claims to be made about regions rather than a massive frontoparietaltemporal network. On that note, to demonstrate the specificity of the effects, we would also like to see a whole-brain searchlight decoding analysis. This would also allow some greater specificity of where in the brain they see effects of the color task (presumably V4), or where in the brain they see effects of object material (e.g., aluminum vs. cork, more on this below). The inclusion of a searchlight analysis would significantly bolster observations.

We did perform the decoding analysis separately in frontal and parietal fROIs, reported in the legend to Figure 3, and did not find a significant effect of hemisphere or frontal or parietal parcel on decoding accuracy.

We agree that a searchlight decoding analysis would be useful, and have run whole-brain searchlight analyses to test location specificity of mass and scenario decoding (Author response image 2).

**Author response image 2. respfig2:** 

Mass decoding in the searchlight procedure was identical to the ROI-based procedure except that the classifier was trained and tested on voxels within searchlight spheres (radius = 3 voxels) rather than individually localized fROIs. The classification accuracy for each sphere was assigned to its central voxel, yielding a single accuracy map in each subject’s native space. We then transformed individual subject maps to a group space (CVS space) and projected data from the volume to the surface. Finally we conducted a one-sample t test over individual subjects’ accuracy maps, comparing accuracy in each voxel to chance. This yielded a group map (shown here) which was assessed at p <0.05 and FWE corrected.

A similar procedure was used in the second searchlight analysis, with scenario decoding tested instead of mass. fROI-based decoding analyses in Experiments 1 and 2 tested the generalization of a neural representation of object mass across multiple physical scenarios: splashing into water, blowing across a flat surface, and falling onto the surface of a pillow. Here we tested where in the brain we see the effects of physical scenario.

As expected, there is a difference between where we see decoding that depends on analyzing patterns of visual information and where we see decoding of underlying physical quantities. The ROI analyses indicate there is decoding in the physics ROIs but the searchlight suggests there is also decoding in adjacent areas in or near motor cortex. Sensitivity to mass in motor regions was not our focus in this work, but it is potentially an interesting finding that would be consistent with physics-based models of motor cortex coding. For instance, the model of Todorov (2000, Nature Neuroscience) posits that central control of the motor periphery is based in part on specifying the forces that must be produced in muscles to achieve voluntary action goals, given properties of muscle physiology and multi-joint mechanics. In general, for any object manipulation task, this will depend on the weight of objects, and so it would not be surprising to see some encoding of object mass in M1. We hope to explore this possibility further in future work.

3) Based on the Experiment 2 design, we are not sure you can make any claims about the automaticity of mass encoding. The study interspersed color and mass blocks, and it is difficult to rule out that the context effect on mass blocks carried over to (and created biases on) color blocks. Moreover, we think it would have strengthened their observations to show that a classifier trained on mass blocks can cross-decode mass on color blocks, rather than just decode mass from the color blocks independently (as was done).

While we did set out with questions about the automaticity of mass encoding, we do agree that we can’t make strong claims about automaticity from the results of Experiment 2 alone, due to the potential of a context effect from the mass task being carried over to the color task. This was, in part, our motivation for designing Experiment 3: to test decoding in an experiment with a physical reasoning task that required implicit understanding of mass, but never explicitly asked about it. From successful decoding during the prediction task in Experiment 3, we conclude that we are not simply decoding a prepared response to the question: “is this object heavy or light,” but an abstract, generalizable representation of mass useful in understanding forces and dynamics and predicting outcomes of physical events. We agree that this result leaves open broader questions of the automaticity of physical representations, e.g., to what extent is the brain’s machinery for physical reasoning running constantly? Which physical dimensions are represented even when they are not useful in a particular task? To what extent is physical reasoning modulated by task and attention? These are fruitful avenues of future work that we now mention in the Discussion section of the paper. Language has been adjusted in the paper to more accurately characterize conclusions and open questions around automaticity.

We did train a classifier on mass blocks and test cross-decoding of mass during the color blocks, and cross-decoding accuracy was not above chance. This suggests that the representation of mass we find is not completely abstract; it generalizes with respect to scenario but not with respect to task, as least in this case where one task does not require physical reasoning. This result is a limit on the automaticity of the mass representation, opening up further questions regarding the degree of abstraction of physical representations, and the types of tasks to which they generalize. We begin to address the automaticity question in Experiment 3: Can a generalizable mass representation be decoded during a physical task that does not ask subjects about mass? This motivation is now clarified at the end of the discussion of Experiment 2. We also note the outstanding questions in the Discussion.

4) The authors said that "A power analysis was used to calculate the appropriate number of subjects for each experiment." We assumed this meant that the authors determine a sample size of 20 was necessary for Experiment 3. Why then did the authors examine mass decoding in the LO/pF/OTC texture areas in only 6 subjects? A fair comparison, to demonstrate equivalent power, would be to demonstrate that object material (e.g., aluminum vs. cork) can be decoded using the same regions (LO/pF/OTC).

As in Experiments 1 and 2, we used a power analysis to calculate the appropriate number of subjects for Experiment 3, which was not 20, but 6. Thus, we began by running 6 subjects in Experiment 3, and significance was obtained. We tested mass decoding in LO/pF/OTC in these initial 6 subjects. It was after these initial 6 subjects that we decided to use eye tracking to ensure subjects were fixating at the center of the screen and to confirm our decoding result was not due to eye movement. We intended to scan 6 subjects with eye tracking to demonstrate equivalent power, but we were required to recruit 2 additional eye tracking subjects as two of the original subjects had eye colors that were too dark for the EyeLink system to successfully track (their brain data was still included in the decoding analysis). Finally, we tested 6 more subjects in a paradigm which included, in addition to the stimuli we describe, additional conditions that tested a different hypothesis. The presentation of the original stimuli remained identical to their presentation to the first 14 subjects. (Specifically, we included additional trials with real-world objects (a cookie, a sock, a key ring, and PVC pipe) with similar masses and coefficients of friction to the cardboard, cork, aluminum and lego cubes in the orthogonalized design. We attempted to cross-decode mass of the cubes on trials with real-world objects, but cross-decoding was not above chance. This may be due to similar limits on generalizability described in Essential revision 3.) In the data reported for Experiment 3 we therefore included all 20 of the subjects run on this experiment, since we had the data, not just the original 6 we planned.

5) Eye Movement Control analysis. This analysis was completely unclear. How are the different measures computed, what time windows were used for analysis (specific events in a trial?), how were the trials collapsed for analysis? No details were provided making it difficult to interpret the authors' claims.

Eye-tracking was used for 6 subjects in Experiment 3 to verify that subjects were fixating on the center of the screen, and to ensure that mass decoding was independent of eye movement. Trials were labeled as light or heavy and low or high friction according to real-world video identity. For each trial, the entire duration of the video (6s) was used for analysis. Mean eye position (deviation from center of the screen) and mean saccade amplitude (averaging over all saccades that occurred in that trial) were calculated for each trial using EyeLink Data Viewer software and MATLAB R2016B. We then used a two-way ANOVA to analyze the interaction between mass and friction and mean eye position and saccade amplitude during the fixation condition and found no significant effects.

6) "A pattern recognition approach to physical reasoning might predict that the neural representations in these regions would hold information about low-level visual features or situation-specific representations of physical variables. In contrast, if these regions support a generalized engine for physical simulation, we would expect to find that they hold representations of abstract physical dimensions that generalize across scenario and other physical dimensions." – We are a bit unconvinced about the logic. Isn't it possible for a pattern recognition technique to implicitly encode these abstract physical dimensions, which are then detected by your decoding approach? For example, suppose we train a reinforcement learning (RL) algorithm in a virtual environment to perform certain tasks. Isn't it possible that physical dimensions are implicitly encoded by the RL algorithm even though the RL algorithm does not utilize an explicit physical generative model? In fact, one might argue that the use of MVPA (linear SVM) in Experiment 1 suggests that the physical representation is implicit, so the authors might simply be decoding the pattern recognition approach's implicit representation of mass (or some other physical attribute) in the brain?

A very interesting question! A pattern recognition approach or model-free RL algorithm, if built on an object-centric representation, could develop an implicit representation of some aspects of unobservable, dynamically relevant physical object properties, such as mass. But, it would only develop representations that approximate the aspects of that physical parameter that are relevant to the particular tasks and physical contexts the algorithm is trained on. It would not be expected to generalize to tasks or contexts different from those in its training. We designed the stimuli in our experiments to test generalization across novel scenarios unseen during training, and find mass representations with meaningful generalization across orthogonal physical dimensions such as friction and motion.

An interesting point of contrast is the paper, “Unsupervised Learning of Latent Physical Properties Using Perception-Prediction Networks” (Zheng et al., 2018), co-authored by one of us (JBT). This paper takes a “pattern recognition”-style approach (i.e., gradient descent in a neural net trained from a million training examples all generated by a conventional physics engine) to learning a dynamics model in the form of an object-centric neural physics engine (defined on a graph-structured neural network). The authors show that they can decode an invariant notion of mass which generalizes to a limited extent: e.g., trained on 6 balls bouncing elastically, it can generalize a decodable mass concept to 3 or 9 balls bouncing elastically. The graph neural net is effectively learning a simulator, at least for these limited cases of balls bouncing: it learns a function that infers the latent physical state of a dynamic scene from past motion history and predicts the next state in time. Thus to the extent that it has a limited mass concept, it does so by virtue of having learned a limited physics engine. Neither the simulator nor the mass concept have been shown to generalize very far beyond the training regimes. But it does show as an existence proof that the concepts of pattern recognition, generative models, physics engines, and simulators are not simple categories that are simply distinct, in the sense that generative models, simulators, and physics engines can in part be learned, and approximations to physically accurate simulators that hold in a limited regime similar to training can be learned by pattern recognition-style training methods. We expect these kinds of models will be interesting to pursue in future work, to see if they can learn more abstract, veridical, invariant notions of physics, but to do that they will likely have to incorporate even more explicit representations of objects and their dynamics.

7) I understand that the authors relied on a network of brain areas that had been identified in a previous study by Fischer et al., 2016, which were shown to be engaged in a physical reasoning paradigm. It does seem, however, from the description of the localizer task, that the contrast of falling direction (i.e., the 'physics' task) vs. color task, would identify a number of different brain areas, not all of which would have similar functions. As one example, intuitively, it seems that the physics task that involved estimating where the leaning tower of blocks would fall would preferentially activate motion-selective areas (i.e., hV5/MT+) – which have been shown in previous fMRI studies to be activated when observers anticipate motion. The group maps illustrated in Figure 3A do indeed suggest an involvement of such motion-selective cortex, especially in the RH. As such, I wondered how meaningful the results are given that there is such a broad network of areas that is being sampled in the analyses. Is the claim that the network as a whole is operating as a 'physics engine', or are more localized regions within the network accounting for this function more so than others? It is also possible that different regions within the network may be more or less likely to show invariance – for example in real-world scenarios when, presumably, more specific rather than invariant representations would be beneficial to support possible actions? Perhaps identifying the areas that were part of this 'network' (either in the figure or a new table) would help readers to make their own assessment.

We agree that the network of areas implicated in physical reasoning by the localizer task is broad and includes regions with general prediction, spatial, and motion functions. The original experiments by Fischer et al., 2016, also found activation of a broad network, including motion-selective areas (MT+), in the physics > color contrast in the same towers localizer used in this study (see Figure 2, Fischer et al., 2016). However, Experiment 2 in Fischer et al. narrowed down the set of fROIs considered candidates for a ‘physics engine’ to a set of bilateral parcels in dorsal premotor cortex and supplementary motor area (Figure 3, Fischer et al., 2016), and bilateral parcels in parietal cortex, which were specifically selective to intuitive physical reasoning. The stimuli in that experiment were pairs of moving dots with motion that implied either social interaction (chasing one another, like in classic animations by Heider and Simmel) or physical interaction (bouncing like billiard balls). After 8s in each video, one of the dots disappeared, and subjects were asked to predict the continuing trajectory of the invisible dot. In the final video frame, the missing dot reappeared, and subjects reported whether it reappeared in the correct location. Both conditions used the same task, requiring mental simulation of spatial paths, but one condition implicitly invoked physical prediction and the other invoked social prediction. The premotor and parietal parcels described above showed significantly greater activation during the physical condition than the social condition, implicating them in the processing of physical information. The remaining fROIs in the broader network showed robust responses to both conditions. We therefore carried out decoding analyses in the set of parcels preferential to physical reasoning in the present study. The role of MT in intuitive physics is an interesting question we are currently investigating in a different study. These points are now discussed in greater detail in the paper.